# Pareto Actor-Critic for Equilibrium Selection in Multi-Agent Reinforcement Learning

**Filippos Christianos** *                                                 *f.christianos@ed.ac.uk*
*University of Edinburgh*

**Georgios Papoudakis** *                                                 *g.papoudakis@ed.ac.uk*
*University of Edinburgh*

**Stefano V. Albrecht**                                                   *s.albrecht@ed.ac.uk*
*University of Edinburgh*

**Reviewed on OpenReview:** *https://openreview.net/forum?id=3AzqYa18ah*

## Abstract

This work focuses on equilibrium selection in no-conflict multi-agent games, where we specifically study the problem of selecting a Pareto-optimal Nash equilibrium among several existing equilibria. It has been shown that many state-of-the-art multi-agent reinforcement learning (MARL) algorithms are prone to converging to Pareto-dominated equilibria due to the uncertainty each agent has about the policy of the other agents during training. To address sub-optimal equilibrium selection, we propose Pareto Actor-Critic (Pareto-AC), which is an actor-critic algorithm that utilises a simple property of no-conflict games (a superset of cooperative games): the Pareto-optimal equilibrium in a no-conflict game maximises the returns of all agents and, therefore, is the preferred outcome for all agents. We evaluate Pareto-AC in a diverse set of multi-agent games and show that it converges to higher episodic returns compared to seven state-of-the-art MARL algorithms and that it successfully converges to a Pareto-optimal equilibrium in a range of matrix games. Finally, we propose PACDCG, a graph neural network extension of Pareto-AC, which is shown to efficiently scale in games with a large number of agents.

## 1 Introduction

Multi-Agent Reinforcement Learning (MARL) in *no-conflict games* (a superset of cooperative games, detailed below) aims to jointly train multiple agents to solve a common task in a shared environment. Recent work has focused on value decomposition (Rashid et al., 2018), communication (Foerster et al., 2016; Wang et al., 2020), agent modelling (Albrecht & Stone, 2018), experience/parameter sharing (Christianos et al., 2020; 2021), and more. However, as we will discuss, many of these approaches do not differentiate between multiple Nash equilibria, and are prone to converging to sub-optimal solutions.

Consider the Stag Hunt game (Fig. 1), a two-agent no-conflict game characterised by a shared set of most-preferred outcomes for all agents (Rapoport, 1966). The Stag Hunt game exhibits two pure-strategy Nash equilibria, one of which is Pareto-optimal and the other is Pareto-dominated, also known as the risk-averse equilibrium (Harsanyi & Selten, 1988).[1] Although the Pareto-optimal equilibrium yields the highest returns for all agents, the risk-averse equilibrium is commonly chosen when there is uncertainty regarding the other agent's actions. In this context, *risk* refers to the possibility of inadvertently selecting a joint action that incurs penalties. The study of Papoudakis et al. (2021) showed that many recent MARL algorithms are

---

*Equal Contribution.

[1] The Stag Hunt game also has one mixed-strategy Nash equilibrium. However, no-conflict games always have *pure-strategy* Pareto-optimal Nash equilibria. For this reason, in the text, we focus our explanations on pure-strategy equilibria.

Agent 2

|  |  | $A$ | $B$ |
|---|---|---|---|
| Agent 1 | $A$ | $(4, 4)$‡ | $(0, 3)$ |
|  | $B$ | $(3, 0)$ | $(2, 2)$† |

Figure 1: Normal-form of the Stag Hunt game. The game has two pure Nash equilibria, one that is Pareto-dominated† (*risk-averse*) and one that is Pareto-optimal‡. In this example, if agent 1 chooses action A, it *risks* receiving zero reward if the other agent chooses B. But, if agent 1 chooses action B it will always receive a reward of at least 2.

unable to converge to the Pareto-optimal equilibrium in the Penalty and Climbing matrix games (both of which share a similar structure to the Stag Hunt example) used by Claus & Boutilier (1998). We corroborate these findings by showing the tendency of MARL algorithms to converge to a risk-averse equilibrium in stateless matrix games. Furthermore, we show that this behaviour can extend to sequential state-based games if they contain sub-optimal equilibria in the form of penalties for failing to coordinate.

The reason that MARL algorithms, such as QMIX (Rashid et al., 2018) or MAPPO (Yu et al., 2021), converge to Pareto-dominated equilibria is rooted in the exploration policies used to explore the state-action space (e.g., using $\epsilon$-greedy or soft policies). For instance, policy gradient algorithms sample actions from the stochastic policy they learn. When the training starts, the stochastic policies are almost uniform, and often dominated by an entropy term used to promote exploration (Mnih et al., 2016). Therefore, during the early stages of training, algorithms that do not consider the joint action value may assign higher expected returns to individual actions that do not require coordination. To illustrate this point, let us consider again the Stag Hunt example where agent 2 is initialised with a uniform policy. In this scenario, the expected reward for agent 1 selecting action A is $4 \times 0.5 + 0 \times 0.5 = 2$, while for action B, it is $3 \times 0.5 + 2 \times 0.5 = 2.5$. Consequently, when applying the policy gradient, agent 1 learns to assign a greater probability to action B. This reinforcement further strengthens the risk-averse (B, B) equilibrium through a positive feedback loop, making action A even less appealing for agent 2.

We propose an algorithm, called *Pareto Actor-Critic (Pareto-AC)*,[2] which is designed to converge to pure-strategy Pareto-optimal Nash equilibria in no-conflict games. The algorithm leverages the characteristic of no-conflict games, where all agents are aware that the Pareto-optimal equilibrium is also preferred by all the other agents, thus avoiding the drawbacks associated with exploring risky joint actions. To do so, the algorithm guides the policy gradient to the Pareto-optimal equilibrium by using a modified advantage estimation that takes into account the no-conflict nature of the studied games, assuming each agent expects others to select actions leading to the same equilibrium. To accomplish its goal, the Pareto-AC algorithm performs computations involving joint actions, which scale exponentially with the number of agents. To address this scalability issue, we also explore the use of tractable graph-based approximations (Böhmer et al., 2020). This approach leads us to propose the *Pareto Actor-Critic with Deep Coordination Graphs (PACDCG)* extension, which aims to attain comparable outcomes (achieving the second-highest returns in the tested games, after Pareto-AC) while mitigating the scalability concern.

In summary, we contribute a novel deep MARL algorithm which i) successfully converges to Pareto-optimal pure Nash equilibria in all 21 distinct $2 \times 2$ no-conflict matrix games (Rapoport, 1966) and three larger matrix-games (Claus & Boutilier, 1998), ii) learns to coordinate in partially observable stochastic games and is the only algorithm (compared to seven state-of-the-art MARL algorithms) that solves all three proposed high dimensional multi-agent tasks, iii) adheres to the Centralised Training Decentralised Execution (CTDE) paradigm that is commonly used in MARL research, and iv) can relax its scaling properties using the proposed PACDCG graph-based approach.

---

[2]Implementation code for Pareto-AC can be found in `https://github.com/uoe-agents/epymarl`.

## 2 Technical Preliminaries

### 2.1 Partially Observable Stochastic Games

We consider a Partially Observable Stochastic Game (POSG) (Hansen et al., 2004) which is defined by $(\mathcal{I}, \mathcal{S}, \mathcal{O}, \mathcal{A}, \Omega, \mathcal{P}, \mathcal{R})$, where $\mathcal{I}$ is the set of $N$ agents $i \in \mathcal{I} = \{1, \ldots, N\}$, $\mathcal{S}$ is the set of states of the game, $\mathcal{A} = A_1 \times \ldots \times A_N$ is the set of joint actions, where $A_i$ is the action set of agent $i$, $\mathcal{O} = \mathcal{O}_1 \times \ldots \times \mathcal{O}_N$ is the joint observation set, where $\mathcal{O}_i$ is the observation set of agent $i$. $\Omega : \mathcal{S} \times \mathcal{A} \times \mathcal{O} \to [0, 1]$ and $\mathcal{P} : \mathcal{S} \times \mathcal{A} \times \mathcal{S} \to [0, 1]$ define probability distributions over the next joint observation and next state, respectively, given the current state and joint action. Finally, $\mathcal{R} : \mathcal{S} \times \mathcal{A} \times \mathcal{S} \to \mathbb{R}^N$ is the reward function that returns a reward $r_i^t$ for each agent $i$ at time step $t$. A solution to a game is a joint policy, $\pi = (\pi_1, \pi_2, \ldots, \pi_N)$, which satisfies certain requirements expressed in terms of the expected return for each agent. The expected return for an agent $i$ under a joint policy $\pi$ and the state transition distribution is given by $G_i(\pi) = \mathbb{E}[\sum_{t=0}^{H-1} \gamma^t r_i^t | \pi_i, \pi_{-i}]$ where $\gamma \in [0, 1]$ is the discount factor, and $H$ is the horizon of the episode. The notation $\pi_{-i}$ is used to refer to the policies of the other agents such that $\pi = (\pi_i, \pi_{-i})$.

In this work, we specifically study *no-conflict games*, in which all agents have the same set of most preferred outcomes. Specifically, in no-conflict games, the agents have the same set of joint policies that maximise their expected returns, formally:

$$\arg\max_{\pi} G_i(\pi) = \arg\max_{\pi} G_j(\pi) \quad \forall i, j \in \mathcal{I} \tag{1}$$

A subset of no-conflict games, which is extensively studied in the community, are the cooperative games in which the reward is common among the agents, i.e. $G_1 = G_2 = \ldots = G_n$ for any joint policy.

### 2.2 Game Theory and MARL Solution Concepts

Game theory and MARL are related topics which study how agents can make strategic decisions. While game theory and MARL use different terminology, in this work we adopt MARL's terminology given the main focus of the paper.

**Definition 1** (Best-Response). *The set of best-response policies for agent $i$ is defined as*

$$BR_i(\pi_{-i}) = \arg\max_{\pi_i} G_i(\pi_i, \pi_{-i}) \tag{2}$$

**Definition 2** (Nash Equilibrium). *A joint policy $\pi$ is a Nash equilibrium if there is no agent that can achieve a higher expected return by unilaterally changing its policy. The Nash equilibrium can also be written using the best-response (BR), such that a joint policy $\pi = (\pi_i, \pi_{-i})$ is a Nash equilibrium if:*

$$\pi_i \in BR_i(\pi_{-i}) \quad \forall i \in \mathcal{I} \tag{3}$$

**Definition 3** (Pareto Optimality). *A joint policy $\pi$ is called Pareto-optimal if no agent can improve its expected return without making the expected return of another agent worse. Formally, a joint policy $\pi$ is Pareto-optimal if:*

$$\nexists \pi' \text{ such that } \forall i : G_i(\pi') \geq G_i(\pi)$$
$$\text{and } \exists i : G_i(\pi') > G_i(\pi) \tag{4}$$

*If such a joint policy $\pi'$ does indeed exist, then it is said that $\pi$ is* Pareto-dominated *by $\pi'$.*

If a joint policy is a Nash equilibrium and Pareto-optimal, we will refer to is as a Pareto-optimal equilibrium.

### 2.3 Policy Gradient and Actor-Critic

Policy gradient methods aim to find the parameters $\phi$ of the parameterised policy $\pi_\phi$ that maximise the expectation, under the parameterised policy, of the discounted episodic sum of rewards. The simplest policy gradient method is REINFORCE (Williams, 1992), which, in single-agent RL, estimates the gradient of the

objective $J(\phi) = \mathbb{E}_\pi [G^t \nabla_\phi \log \pi_\phi(a^t|o^{:t})]$, where $G^t$ is the Monte-Carlo return estimation, and $o^{:t}$ the history of observations. To minimise the variance of the gradient estimation in REINFORCE, it is common to (i) subtract the state value from the returns, and (ii) approximate the returns using the state value function. Therefore, there is also the need to learn a function that approximates the state value function $V_\theta$ with parameters $\theta$. The class of algorithms that learn both a policy and a value function is called actor-critic. A commonly used actor-critic algorithm is advantage actor-critic (A2C) (Mnih et al., 2016). An application of A2C in MARL is the Independent A2C (IA2C) algorithm, where each agent is trained by using A2C and ignoring the presence of the other agents in the environment, with the actor of agent $i$ minimising:

$$\mathcal{L}(\phi_i) = -\mathbb{E}_{a_i^t \sim \pi_i, a_{-i}^t \sim \pi_{-i}}[\log \pi(a_i^t|o_i^{:t}; \phi_i)(r_i^t + \gamma V(o_i^{:t+1}; \theta_i) - V(o_i^{:t}; \theta_i))], \tag{5}$$

and the critic of agent $i$ minimising:

$$\mathcal{L}(\theta_i) = \mathbb{E}_{a_i^t \sim \pi_i, a_{-i}^t \sim \pi_{-i}}[(r_i^t + \gamma V(o_i^{:t+1}; \theta_i) - V(o_i^{:t}; \theta_i))^2]. \tag{6}$$

*Centralised Training Decentralised Execution (CTDE)* is a multi-agent paradigm that involves the sharing of information among agents during training, while ensuring that once the training is complete, agents can execute their policies based solely on their own observations. Various CTDE extensions of independent actor-critic methods have been proposed in the literature (Lowe et al., 2017; Foerster et al., 2018), wherein the critic is conditioned on the joint trajectory of all agents to enhance the approximation of the expected return for each agent. The centralised training variant of Independent Advantage Actor-Critic (IA2C), commonly referred to as MAA2C (Papoudakis et al., 2021) or Central-V (Foerster et al., 2018) in the literature, is trained to minimise the following loss function for the actor:

$$\mathcal{L}(\phi_i) = -\mathbb{E}_{a_i^t \sim \pi_i, a_{-i}^t \sim \pi_{-i}}[\log \pi(a_i^t|o_i^{:t})(r_i^t + \gamma V(s^{t+1}) - V(s^t))], \tag{7}$$

and for the critic:

$$\mathcal{L}(\theta_i) = \mathbb{E}_{a_i^t \sim \pi_i, a_{-i}^t \sim \pi_{-i}}[(r_i^t + \gamma V(s^{t+1}) - V(s^t))^2], \tag{8}$$

for each agent $i$. The critic in MAA2C utilises the fully observable state $s_t$, which can also be approximated as the concatenation of observations from all agents. Recently, Lyu et al. (2023) discussed how centralised critics conditioned on the state can introduce bias and variance to their estimation. However, this concern can be alleviated by ensuring that the state $s^t$ includes all the information of the history of observations $o^{1:t}$ or by using recurrent critics (i.e. the critic being conditioned on $s^{1:t}$). Throughout the document, we will adopt the notation of using state $s^t$ as an input to the critic networks, while the actor networks will take the individual agent history of observations $o^{:t}$ as inputs, thus adhering to the principles of CTDE.

## 3 Pareto Actor-Critic

### 3.1 Optimisation Objective

Revisiting the Stag Hunt game of Fig. 1, we observe that agents can reach the desired Pareto-optimal equilibrium if they coordinate. If we assume that it is known by both agent that the other agent is not going to deviate from this equilibrium, then they can both select action $A$. This can be formally understood by the nature of no-conflict games (Eq. 1) and that the joint policy that maximises the expected return of one agent, maximises the returns of all agents as well.

We make use of this property of no-conflict games in Pareto-AC while preserving the decentralised interaction of each agent with the environment. For a policy $\pi_i$ of agent $i$, let $\pi_{-i}^+$ be in the set of policies of other agents $-i$ that maximise the returns of agent $i$:

$$\pi_{-i}^+ \in \arg\max_{\pi_{-i}} G_i(\pi_i, \pi_{-i}). \tag{9}$$

Learning an agent policy $\pi_i$ by maximising $G_i$, given that the other agents follow $\pi_{-i}^+$, leads to a joint policy $\pi^+ = (\pi_i, \pi_{-i}^+)$ that maximises the returns of agent $i$. However, from the definition of no-conflict games, that

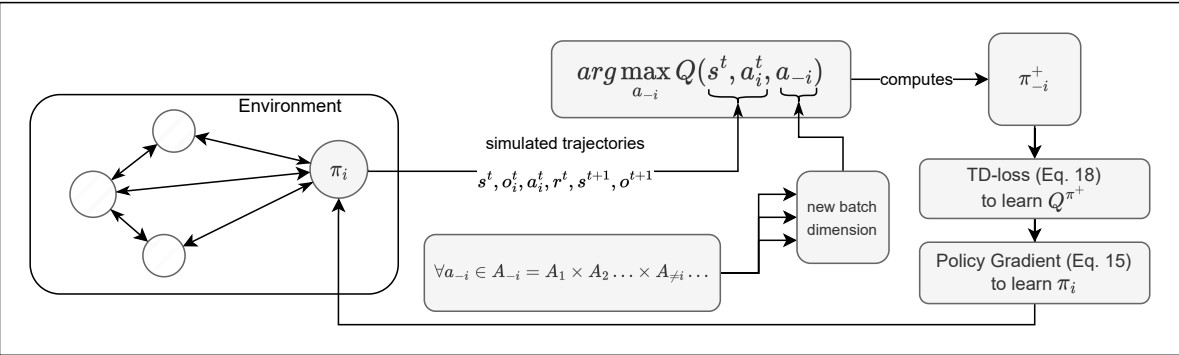

Figure 2: High-level overview of Pareto-AC. Trajectories are collected by simulating agents in an environment (left). The policy $\pi^+_{-i}$ is approximated by evaluating all the joint actions of other agents (i.e. all $a_{-i} \in A_{-i}$) with the argmax operator on the joint Q value. $\pi^+_{-i}$ can be used to train $Q^{\pi^+}$ (Eq. 18), which in turn can be used in the policy gradient of Pareto-AC (Eq. 15). Pareto-AC trains all agents using the same algorithm.

is also the joint policy that maximises the returns of any other agent in $\mathcal{I}$. Pareto-AC learns such a policy for each agent, replacing the typical Nash equilibrium solution (Eq. 3) with:

$$\pi_i \in \arg\max_{\pi_i} G_i(\pi_i, \pi^+_{-i}). \tag{10}$$

In practice, during exploration the trajectories are gathered using the joint policy $\pi = (\pi_i, \pi_{-i})$, while we optimise the joint policy $\pi^+ = (\pi_i, \pi^+_{-i})$. A high-level outline of Pareto-AC is presented in Fig. 2, while individual components are presented below. This is the standard settings of off-policy actor-critic (Degris et al., 2012). We propose a similar objective in our settings:

$$J(\phi_i) = \sum_{s \in \mathcal{S}} d^\pi(s)[V_i^{\pi^+}(s)], \tag{11}$$

where $V_i^{\pi^+}$ is the state value function of agent $i$ under the joint policy $\pi^+ = (\pi_i, \pi^+_{-i})$, and $d^\pi$ is the on-policy distribution of the environment's states under the joint policy $\pi = (\pi_i, \pi_{-i})$.

### 3.2 Defining the Pareto Actor-Critic Algorithm

Our proposed algorithm aims to optimise the objective given in Equation (11) using an off-policy actor-critic method. Note that during training one would not have samples from the state distribution $d^{\pi^+}$, but from the state distribution $d^\pi$ (recall that $\pi$ is the joint policy). Building on the work of Degris et al. (2012), we have that:

$$
\begin{aligned}
\nabla J(\phi_i) &\approx \sum_s d^\pi(s) \sum_a \nabla \pi^+(a|s) Q_i^{\pi^+}(s, a) \\
&= \sum_s d^\pi(s) \sum_{a_i, a_{-i}} \pi^+_{-i}(a_{-i}|s, a_i) \nabla \pi_i(a_i|s) Q_i^{\pi^+}(s, a_i, a_{-i}) \\
&= \sum_s d^\pi(s) \sum_{a_i, a_{-i}} \pi_i(a_i|s) \pi^+_{-i}(a_{-i}|s, a_i) \nabla \log \pi_i(a_i|s) Q_i^{\pi^+}(s, a_i, a_{-i}).
\end{aligned}
\tag{12}
$$

The approximately equal sign ($\approx$) is included because the gradient of the Q-value $Q_i^{\pi^+}$ with respect to the parameters is ignored. In their Theorem 1 and 2, Degris et al. (2012) have shown that following the estimated gradient of Equation (12), assuming a sufficiently small learning rate and that the policy is represented in a tabular form, results in improving the objective. Assuming the two joint policies $\pi_{old}$ and $\pi^+_{old}$ before the update, and after $\pi_{new}$ and $\pi^+_{new}$, it stands that:

$$\sum_s d^{\pi_{old}}(s)[V_i^{\pi^+_{new}}(s)] \geq \sum_s d^{\pi_{old}}(s)[V_i^{\pi^+_{old}}(s)]. \tag{13}$$

$$\begin{array}{cc} & \pi_2 = \mathcal{U} \\ \text{Agent 1} \quad \begin{array}{c} A \\ B \end{array} & \begin{array}{|c|} \hline 2 \\ \hline 2.5 \\ \hline \end{array} \end{array} \qquad\qquad \begin{array}{cc} & \pi_2^+ \\ \text{Agent 1} \quad \begin{array}{c} A \\ B \end{array} & \begin{array}{|c|} \hline 4 \\ \hline 3 \\ \hline \end{array} \end{array}$$

(a) Expected reward for agent 1 in the Stag Hunt game given that agent 2 follows a uniformly random policy $\pi_2(A) = 0.5$ and $\pi_2(B) = 0.5$. While the Pareto-optimal joint action is (A,A), agent 1 would expect a higher reward if it selects action B. This further reinforces agent 2 to also prefer action B leading to the sub-optimal equilibrium (B,B).

(b) Expected reward for agent 1 in the Stag Hunt game given that agent 2 follows the $\pi_2^+$, where $\pi_2^+(A) = 1.0$ and $\pi_2^+(B) = 0$. If agent 2 follows policy $\pi_2^+$, agent 1 will prefer action $A$. This will lead agent 2 to prefer action $A$ too, which will lead to the Pareto-optimal joint-action $(A, A)$.

Figure 3: Expected return for agent 1 under the expectation that agent 2 is using a uniform policy (in a), and the policy $\pi^+$ (in b).

Additionally, the gradient update results in increasing the state value function for every state:

$$V_i^{\pi_{new}^+}(s) \geq V_i^{\pi_{old}^+}(s) \quad \forall s \in \mathcal{S} \tag{14}$$

Below, we rewrite the optimisation objective as a loss function, where the actor is conditioned on its individual history of observations. Additionally, we subtract the state value function as a baseline to reduce the variance:

$$\mathcal{L}(\phi_i) = -\mathbb{E}_{s \sim d^\pi, a_i^t \sim \pi_i, a_{-i}^t \sim \pi_{-i}^+}[\log \pi(a_i^t | o_i^{:t})(Q^{\pi^+}(s^t, a_i^t, a_{-i}^t) - V^{\pi^+}(s^t))]. \tag{15}$$

Equation (15) resembles Equation (7) of the MAA2C algorithm, but instead samples the actions of other agents from $\pi_{-i}^+$. In the rest of the paper, the state value and state-action value functions under policy $\pi^+$ will be abbreviated as $V$ and $Q$ respectively. We also replace $r_i^t + \gamma V(s_i^{t+1}; \theta_i)$ with the equivalent $Q(s^t, a_i^t, a_{-i}^t)$ in Equation (15), deviating from typical actor-critic implementations. Additionally, we train a separate network to approximate the state value that is used as baseline in the policy gradient estimation. As a side note, one could avoid the need of training a separate state value network, by computing the baseline as:

$$V(s^t; \theta_i) = \sum_{a_i} \pi(a_i^t | o_i^t; \phi_i) Q(s^t, a_i^t, a_{-i}^{+,t}). \tag{16}$$

Foerster et al. (2018) have shown that the baseline described in Equation (16) does not add bias to the policy gradient estimation. Alternatively, we found that training a separate state value results in slightly higher returns. The loss, described in Eq. (15) above, maximises the Q-values but *ignores* any miscoordination with the rest of the agents, as it assumes the use of $\pi^+$. While this loss still requires centralised training, the actor can be used independently since it is only conditioned on the history of local observations of each agent. We characterise Pareto-AC as an on-policy algorithm, because from the point of view of each agent $i$ its local trajectory is on-policy. However, the joint trajectories are off-policy because of the assumption that the other agents $-i$ follow the policy $\pi_{-i}^+$. The pseudocode of Pareto-AC is presented in Algorithm 1.

The Pareto-AC algorithm is designed to enable agents to learn policies that achieve Pareto-optimality in no-conflict games. We now provide an intuition for why the Pareto-AC algorithm is effective, using the example of a Stag Hunt game (Fig. 1). In the Stag Hunt game, directly following the policy gradient tends to converge to a sub-optimal equilibrium. This occurs because initially, agents have policies that are close to uniform. In the case where agent 2 has a uniform policy, the expected returns for the Stag Hunt actions A and B of agent 1 are 2 and 2.5, respectively. This reinforces the selection of action B, making it more likely for agent 1 to choose it. Consequently, agent 2 also tends to prefer action B, resulting in the (B, B) joint action. This example, along with the expected returns, is depicted in Fig. 3a.

In contrast, Pareto-AC alters these expectations for any possible policy of the other agents, even if these other policies heavily favour a sub-optimal action. In the Stag Hunt game, the expected rewards of agent 1 for action A and B become 4 and 3, respectively, irrespective of the policy $\pi_{-i}$ that is actually followed by the other agents $-i$. This is illustrated in Fig. 3b.

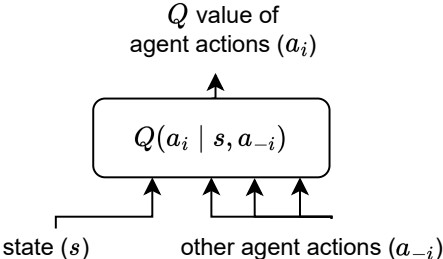

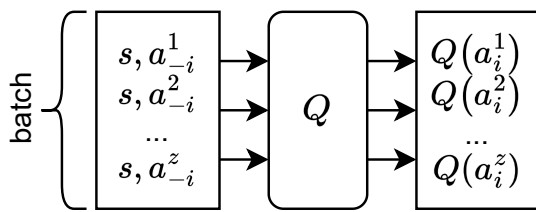

(a) The $Q$ network receives as inputs the state and the actions of the other agents (concatenated together) and outputs a $Q$ value for each of the discrete actions of agent $i$.

(b) The $Q$ network architecture allows us to compute all the combinations of other agent actions *in parallel*, in a single forward operation. $z$ is the number of joint actions $a_{-i}$ for other agents.

Figure 4: Pareto-AC Q network architecture and parallel computation.

The distinction between the standard policy gradient and Pareto-AC becomes evident when comparing the two scenarios: in the former case, agents learns to prefer action B, while in the latter case, the agents prefer action A, leading to the Pareto-optimal equilibrium. These examples extend beyond simple matrix games and Pareto-AC can guide the agents towards learning policies that achieve Pareto-optimality in multi-agent environments.

### 3.3 Computing $\pi^+_{-i}$ Using a Joint State-Action Value Network

This section addresses the computation of $\pi^+_{-i}$. The policy $\pi^+_{-i}$ of other agents can be approximated at any timestep $t$ using a joint action value network (also known as a critic):

$$\pi^+_{-i} \in \arg\max_{a_{-i}} Q(s^t, a_i^t, a_{-i}) \tag{17}$$

The critic estimates the Q-value over the joint actions and can calculate the maximum Q-value over the other agents $-i$ from the perspective of agent $i$, i.e. $\max_{a_{-i}} Q(s^t, a_i^t, a_{-i}^t)$. The max-operator is the core of our approach. Calculating the maximum Q-value over the possible actions of the other agents translates to *assuming* that the other agents will follow $\pi^+_{-i}$. Therefore, from the point of view of each agent, the other agents will execute any actions required to reach the Pareto-optimal equilibrium.

The number of joint actions scales exponentially in the number of agents in the environment, rendering the computation of the Q-values of all joint actions computationally expensive. To allow for fast computation of the Q-value of all joint actions, our algorithm includes two distinct design choices (also seen in Figures 4a and 4b).

First, we adopt a strategy that avoids the need for an exponentially increasing critic-network output. We modify the architecture by incorporating the actions of the other agents as inputs (as illustrated in Fig. 4a). By doing so, the size of the network scales linearly with the number of agents, effectively mitigating the computational burden (Foerster et al., 2018).

Second, we parallelise the computation of $Q(s^t, \cdot, a_{-i}^t)$. We do so by devising a critic that accepts a batched input that contains the observation concatenated with all the possible $a_{-i}$. The combinations of the actions $a_{-i}$ is the Cartesian product of the action spaces $A_{-i} = \times_{j \neq i} A^j$ which can then be concatenated with the joint observation of all agents. By utilising this input for the critic-network, one is able to compute all the corresponding Q-values in a single forward pass, as depicted in Fig. 4b.

The critic network can be trained by following the methodology above. Note that in contrast to common on-policy actor-critic architectures (e.g., A2C and PPO), Pareto-AC's critic approximates the Q-values instead of the state values. Training the Q-function is done by minimising the TD-error between the selected actions

and the target, which includes the described max-operator:

$$\mathcal{L}(\theta_i) = \mathbb{E}_{a_i^t, a_i^{t+1} \sim \pi_i, a_{-i}^t \sim \pi_{-i}}[(r_i^t + \gamma \max_{a_{-i}} \hat{Q}(s^{t+1}, a_i^{t+1}, a_{-i}) - Q(s^t, a_i^t, a_{-i}^t))^2] \tag{18}$$

with $\hat{Q}$ being the target network (updated either with hard or soft updates).

To ensure that the learned Q-value can be used in place of the expected returns in Eq. (15), SARSA-style (Rummery & Niranjan, 1994) updates are used for estimating the Q-value. The use of SARSA allows the estimated Q-value for agent $i$ to be the expected returns under the policy that is currently used by agent $i$. However, employing one-step SARSA tends to result in an overestimation of the returns. Therefore, we adopt N-step SARSA for training the critic. An ablation study with respect to the number of steps is presented in Section 4.5, where the need for employing the N-step computation is highlighted.

---

**Algorithm 1** Pseudocode of Pareto-AC algorithm from the point of view of agent $i$.

---

    **Input**: Randomly initialised parameters of the actor $\phi_i$ and critic $\theta_i$ of agent $i$.
    **Input**: The desired number of maximum training steps $N_{\max}$.
    **Input**: A process that can interact the environment.
    $num_{\text{episodes}} \leftarrow 0$
    **while** $num_{\text{episodes}} < N_{\max}$ **do**
        Create $M$ parallel environments from the environment simulator
        **for** $t = 0, ..., H - 1$ **do**
            **for** every environment $m$ in $M$ **do**
                Get observations $o_i^t$ and $o_{-i}^t$
                Sample action $a_i^t \sim \pi_i(a_i^t | o_i^t)$ and $a_{-i}^t \sim \pi_{-i}(a_{-i}^t | o_{-i}^t)$
                Perform the actions and get $o_i^{t+1}, o_{-i}^{t+1}, r_i^{t+1}$
            Gather the sequences of all $M$ environments in a single batch $B$
            Compute a policy $\pi_{-i}^+ \in \arg\max_{a_{-i}} Q(s^t, a_i^t, a_{-i})$ for all $t$
            Update the critic using Equation (18)
            Update the actor using Equation (15)
            $num_{\text{episodes}} \leftarrow num_{\text{episodes}} + M$
    **Outputs**: Trained parameters of the actor $\phi_i$ and the critic $\theta_i$ of agent $i$

---

### 3.4 Addressing the Exponential Cost of Calculating $\pi_{-i}^+$

The architecture proposed in Section 3.3 allows for efficient computation of Q-values up to a small number of agents. However, the computation becomes prohibitively expensive with a larger number of agents, requiring a batch size that scales exponentially with the number of agents. Value Function Factorisation (Crites & Barto, 1998; Guestrin et al., 2001) literature includes many proposals to enable efficient scaling to more

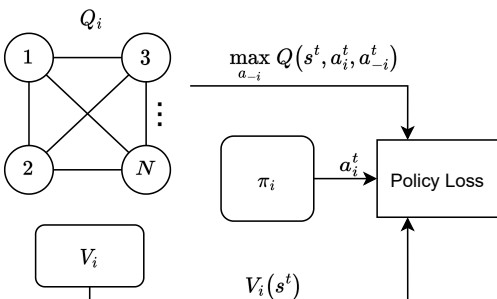

Figure 5: Architecture of the PACDCG algorithm. The critic computes the maximum Q-value over the joint actions.

agents. For example, using coordination graphs (Kok & Vlassis, 2005; Van der Pol & Oliehoek, 2016) one can approximate the max Q-values via a graph-based representation. In this work, we specifically build on top of Deep Coordination Graphs (DCG) (Böhmer et al., 2020), an algorithm that calculates the max Q-values in a Centralised Training *Centralised Execution* approach.

Equations (15) and (18) only require the maximum Q-value of the joint actions, as well as the Q-values of the specific joint action that is executed at each time step and not the Q-values of all joint actions. DCG assumes that the Q-value of a joint action can be factorised using a fully-connected undirected graph, where each node represents an agent that has its own Q-value. Each edge between two nodes represents the pairwise Q-value between the two agents. Both the individual as well as the joint Q-values are approximated using neural networks. Consider a graph $G = (\mathcal{V}, \mathcal{E})$, where $\mathcal{V}$ is the set of nodes, and $\mathcal{E}$ is the set of edges. Assuming that $Q_i$ is the individual Q-value of agent $i$, and $Q_{ij}$ is the pairwise Q-value of agents $i$ and $j$, following the works of Kok & Vlassis (2006); Castellini et al. (2019), and more specifically in our case the coordination graph as proposed by Böhmer et al. (2020), the joint Q-value can be computed as:

$$Q(s^t, a^t) = \sum_i Q_i(s^t, a_i^t) + \sum_{ij} Q_{ij}(s^t, a_i^t, a_j^t) \tag{19}$$

DCG is trained using TD-learning to approximate the joint Q-value. As Böhmer et al. (2020) note, DCG is an extension of VDN, with the main difference between the two of them being that DCG also models the pairwise utilities between the agents. If the edges are removed, DCG collapses to VDN. Böhmer et al. (2020) use a message-passing algorithm to estimate the pairwise messages and the individual actions that maximise Q-value over the joint actions. Using DCG as critic we can compute the action that maximise the joint Q-value with complexity $\mathcal{O}(k|A|(|A|+N)\mathcal{E})$ (Böhmer et al., 2020), where $k$ is the number of message passing iterations, $|A|$ the size of the largest single-agent action space, and $\mathcal{E}$ the number of edges, in contrast to computing the Q-values of all joint actions with complexity $\mathcal{O}(|A|^N)$. A difference between Pareto-AC and PACDCG critics is that in PACDCG the critic estimates that the other agents $-i$ perform the action that maximises the joint Q-value irrespective of the action that agent $i$ performed, while in Pareto-AC the critic estimates that the other agents $-i$ perform the action that maximises the joint Q-value while the action of the agent $i$ is fixed, i.e. $\pi_{-i}^+$. A variation of Pareto-AC that computes the same values as the PACDCG critic shows worse performance (see motivation of Pareto-AC in Section 3.1). Additionally, we use a separate neural network to approximate a centralised state-value function. Figure 5 presents the architecture of the PACDCG algorithm. Note that PACDCG is restricted to cooperative environments, as a direct inheritance of this requirement from the DCG algorithm.

## 4    Experiments

In this section, we experimentally evaluate Pareto-AC and PACDCG in several multi-agent games and compare them against related baselines. We evaluate whether Pareto-AC and PACDCG achieve superior returns compared to the baselines as well as converge to a Pareto-optimal equilibrium.

### 4.1    Multi-Agent Games

**Matrix Games:** Three common-reward multi-agent matrix games proposed by Claus & Boutilier (1998): the Climbing game with two and three agents and the Penalty game. The payoff matrices of the Climbing game with two agents and the Penalty game are presented in Figure 6. The Climbing game is a two agents matrix game with three actions per agent. It has in total seven Nash equilibria (two pure-strategy and five mixed-strategy), with the joint action $(A, A)$ being the Pareto-optimal one. We also extend the Climbing game to three agents, with one Pareto-optimal joint action $(A, A, A)$. The Penalty game is a two agents matrix game with three actions per agent. It has in total five Nash equilibria (three of which are pure-strategy), with Pareto-optimal Nash equilibria being the joint actions $(A, C)$ and $(C, A)$. These matrix games have one or more Pareto-dominated Nash equilibria and one or more Pareto-optimal Nash equilibria.

**Boulder Push:** In the Boulder Push game (illustrated in Figure 8a), two agents and a boulder are situated within an $8 \times 8$ grid-world. Boulder Push shares similarities with Box-Pushing (Seuken & Zilberstein, 2007)

**Climbing game (left)**

|  | Agent 2 A | Agent 2 B | Agent 2 C |
|---|---|---|---|
| **Agent 1 A** | $11^{\ddagger}$ | $-30$ | $0$ |
| **Agent 1 B** | $-30$ | $7^{\dagger}$ | $0$ |
| **Agent 1 C** | $0$ | $6$ | $5$ |

**Penalty game (right)**

|  | Agent 2 A | Agent 2 B | Agent 2 C |
|---|---|---|---|
| **Agent 1 A** | $-100$ | $0$ | $10^{\ddagger}$ |
| **Agent 1 B** | $0$ | $2^{\dagger}$ | $0$ |
| **Agent 1 C** | $10^{\ddagger}$ | $0$ | $-100$ |

Figure 6: Normal-form of the Climbing (on the left), and the Penalty (on the right) games. The Climbing game has two pure Nash equilibria, where the Pareto-optimal$^{\ddagger}$ equilibrium is (A,A). The Penalty game has three pure Nash equilibria, where the Pareto-optimal$^{\ddagger}$ equilibria are the (A,C) and (C,A).

|  | Agent 3 A | | | Agent 3 B | | | Agent 3 C | | |
|---|---|---|---|---|---|---|---|---|---|
|  | Agent 2 A | Agent 2 B | Agent 2 C | Agent 2 A | Agent 2 B | Agent 2 C | Agent 2 A | Agent 2 B | Agent 2 C |
| **Agent 1 A** | $11^{\ddagger}$ | -30 | 0 | -30 | 0 | 0 | -30 | 0 | 0 |
| **Agent 1 B** | -30 | 0 | 0 | 0 | $7^{\dagger}$ | 0 | 0 | 0 | 0 |
| **Agent 1 C** | 0 | 0 | 0 | 0 | 0 | 0 | 0 | $6^{\dagger}$ | 5 |

Figure 7: Normal-form of the Climbing game with three agents. The game has two pure Nash equilibria, where the Pareto-optimal$^{\ddagger}$ equilibrium is $(A, A, A)$.

but emphasises penalties for agent miscoordination. Each agent possesses four available actions for cardinal movement directions and observes relative distances to other agents, the boulder, and grid boundaries. Successful boulder movement requires all agents to move in unison, aligning with arrows shown in Fig. 8a while occupying highlighted positions. Agents are rewarded with 0.1 for successfully contributing to boulder movement. Conversely, if agents attempt (and fail) to push the boulder independently, they incur a penalty of $-0.1$. The optimal joint policy is estimated to yield expected returns of 1.35 for each agent. This calculation comprises a reward of 1 for task completion and an additional 0.35 for participating on average approximately 3.5 times in boulder pushes. Episodes conclude either when the boulder reaches the grid's end or after a set duration of 50 time steps.

**Level-Based Foraging (LBF):** In this game, one food item is placed in a 5x5 grid world (Christianos et al., 2020; Papoudakis et al., 2021), as depicted in Figure 8b. At the start of each episode, both agents and the food item are assigned levels. The agents' primary objective is to forage the food item. To do so, a group of agents can successfully forage the food item only if the sum of their levels is equal to or greater than the level of the food item. Each agent's observation includes their level, as well as the levels and relative distances to other agents and the food item. Agents can move in the four cardinal directions, attempt to forage the food or take no action. Successful coordination and foraging yield a reward of +1 for all participating agents, while failed attempts result in a penalty of -0.6 for each agent. Episodes end when either the food item is successfully gathered or after 25 time steps. The LBF task has two versions: one with two agents and one with three agents. In the two-agent LBF, both agents must simultaneously forage the food item. In the three-agent LBF, the food item's level is randomly initialised, meaning it may require one, two, or all three agents to forage it at once.

## 4.2 Experiment Details

We compare the average evaluation returns achieved by Pareto-AC against the evaluation returns achieved by seven MARL algorithms: MAPPO (Yu et al., 2021), MAA2C, VDN (Sunehag et al., 2018), QMIX (Rashid et al., 2018), QTRAN (Son et al., 2019), QPLEX (Wang et al., 2021), and Friends Q-Learning (Friends-QL)(Littman, 2001). Table 1 summarises all evaluated algorithms and their core properties.

Friends-QL was proposed in a tabular form by Littman (2001). In this work, the architecture of Figs. 4a and 4b was used to apply it in the deep learning setting. With Friends-QL, each agent consists of a neural network that approximates the joint Q-value. During execution, each agent selects its action assuming that

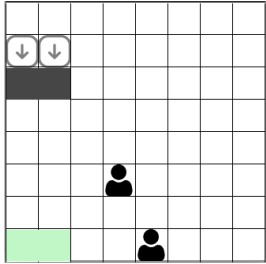

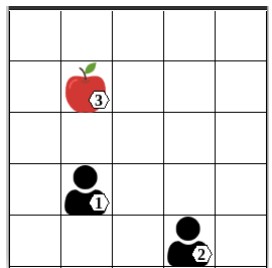

(a) Boulder Push environment.        (b) Level-Based Foraging environment.

Figure 8: Boulder Push and the Level-Based Foraging environments. In both environments, the agents are penalised if they individually try to complete the task without cooperating with the other agent.

Table 1: Overview of the evaluated algorithms.

|  | Value Network | On-/Off- Policy | Environment Settings |
|---|---|---|---|
| Pareto-AC (Ours) | Joint-Action | On | No-Conflict |
| PACDCG (Ours) | Joint-Decomposition | On | Cooperative |
| Friends-QL | Joint-Action | Off | No-Conflict |
| MAA2C | State Value | On | All |
| MAPPO | State Value | On | All |
| VDN | Joint-Decomposition | Off | Cooperative |
| QMIX | Joint-Decomposition | Off | Cooperative |
| QTRAN | Joint-Decomposition | Off | Cooperative |
| QPLEX | Joint-Decomposition | Off | Cooperative |

other agents will choose the action that maximises the joint Q-value. Friends-QL is the Temporal Difference equivalent of Pareto-AC, where both algorithms assume that the other agents will follow $\pi^+_{-i}$, with the different being that that Pareto-AC uses an actor-critic method to learn the policy while Friends-QL uses Q-Learning.

All on-policy algorithms use N-step returns to train their critics. In the case of Pareto-AC, and PACDCG, the use of N-step returns can render the Q-value estimation to be under a different policy from $\pi^+$ for the first N steps. While this is not theoretically justified, we found it a necessary modification to reduce overestimation as we experimentally show in Section 4.5.

Gradient updates are performed at the end of each episode. Value-based algorithms (VDN, QMIX, QTRAN, QPLEX, and Friends-QL) use an experience replay to break the correlation between consecutive samples, and the gradient updates are performed by uniformly sampling a batch of 32 episodes. Actor-critic algorithms (MAA2C, MAPPO, Pareto-AC, PACDCG) use ten parallel processes to break the correlation between consecutive samples, and the gradient updates are performed using a batch of the ten aggregated episodes. Following the work of Papoudakis et al. (2021), we retain the same number of updates across all algorithms, meaning that on-policy algorithms observe 10 times more environment timesteps compared to off-policy.

All neural networks consist of two fully connected layers in fully-observable environments (matrix games and LBF), while the first layer of the actor is a GRU in the Boulder Push environment. The parameters of all networks are optimised using the Adam optimiser (Kingma & Ba, 2015). There is no parameter sharing among the agents. Pareto-AC and PACDCG were implemented based on the EPyMARL codebase (Papoudakis et al., 2021). The implementation of PACDCG's critic was based on the official implementation of DCG (Böhmer et al., 2020).

We performed a hyperparameter search to systematically examine multiple configurations for the training process for *both the baseline algorithms and Pareto-AC*. Our approach ensured fairness by maintaining a

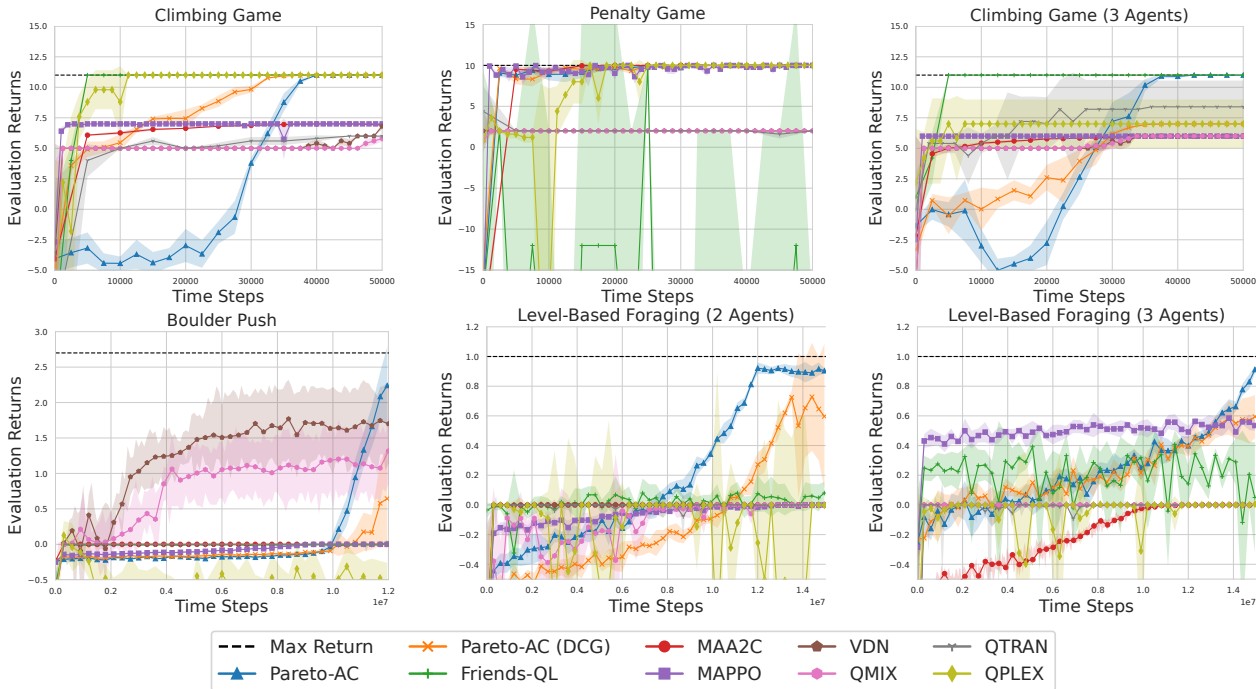

Figure 9: Average episodic evaluation returns and 95% confidence interval over 5 seeds of Pareto-AC and the evaluated baselines in six multi-agent environments.

roughly equal number of search configurations for all algorithms under consideration. A detailed description of the hyperparameters can be found in the appendix.

### 4.3 Average Evaluation Returns

Figure 9 presents the average evaluation returns over five different seeds, which are achieved by Pareto-AC and four evaluated baselines in the aforementioned environments. Figure 10 illustrates the same runs by summarising the maximum performance achieved by each algorithm-environment pair.

**Matrix Games:** Pareto-AC is the only method that converges to the Pareto-optimal equilibria in all games. Friends-QL converges to the Pareto-optimal equilibrium in the Climbing games with both two and three agents, but it does not converge in the Penalty game, to any of the equilibria. The reason behind this can be attributed to the fact that the Penalty game has two Pareto-optimal equilibria. In Friends-QL the agents do not have an explicit mechanism for deciding their joint action and are solely choose their actions based on the fact that the other agent will choose the action that maximise the joint Q-values. By looking into the individual runs and evaluations of Friends-QL in the Penalty game, we observe that in all evaluation points the agents either select one of the Pareto-optimal equilibria or choose the joint actions $(A, A)$ or $(C, C)$ which result in $-100$ penalty. This is a direct result of miscoordination between the agents selecting different Pareto-optimal equilibria. Pareto-AC does not have this problem, as during the training process if agent 1 assigns a higher probability say to action A (e.g. 0.51), agent 2 will be strongly reinforced to converge to the respective action (C in this case) and avoid the heavily-penalised A action (and the opposite direction also holds). MAA2C and MAPPO both converge to the Pareto-optimal equilibrium in the Penalty game. However, both of them converge a sub-optimal joint action in the two Climbing games. PACDCG and QPLEX converge to the Pareto-optimal equilibria in the Climbing game with two agents, and the Penalty game. PACDCG converges to a Pareto-dominated equilibrium in the Climbing game with three agents. We believe that this is due to the graph factorisation. The DCG critic only models pairwise value functions, and is not able to represent values in environments where the reward needs the cooperation of more than two agents.

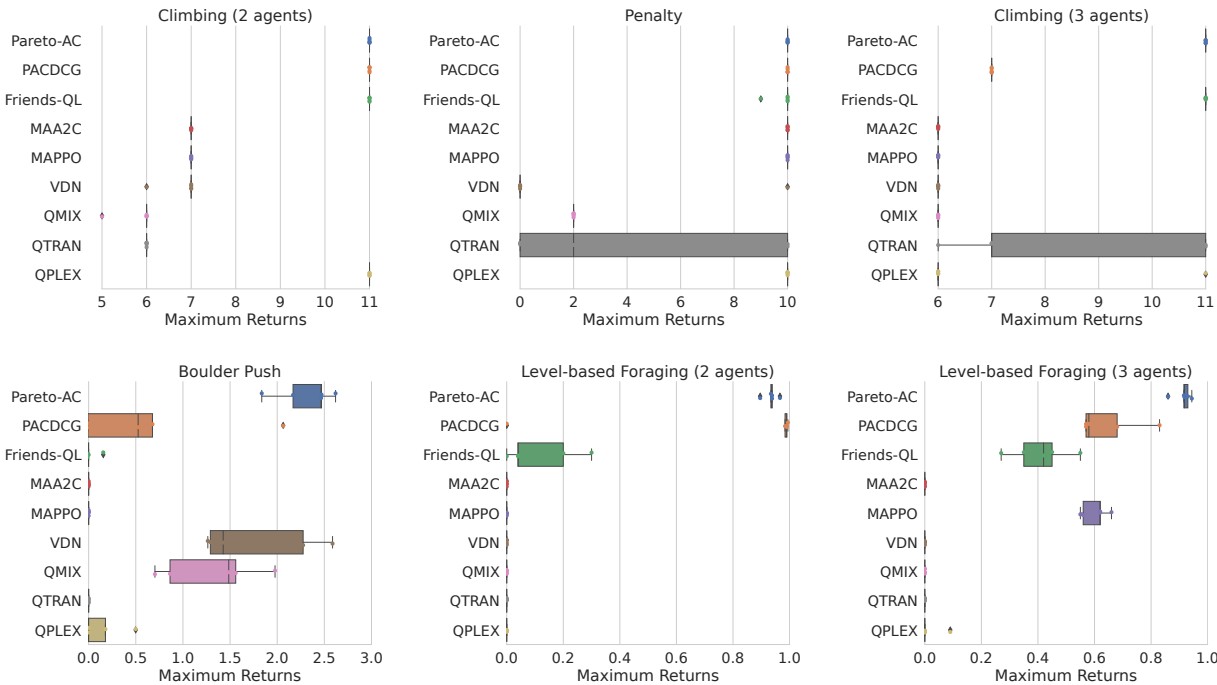

Figure 10: Box plot with maximum evaluation returns for Pareto-AC and the evaluated baselines in six multi-agent environments. The box plot draws a box from the first to the third quartile, excluding outliers and includes a line that goes through the box at the median. Each individual dot signifies the maximum returns achieved by a single run (with a distinct seed).

**Boulder Push:** Pareto-AC converges to higher returns compared to all the other baselines. During the first half of the training, Pareto-AC is receiving negative returns which is evidence to support that it insists on exploring promising joint actions even if they are initially penalised. VDN and QMIX also learn to complete the task, however, not as successfully as Pareto-AC. This indicates that while the agents learn to coordinate in some case, there are time steps where the agents miscoordinate and this results in getting penalised. Finally, PACDCG achieves returns that are slightly higher than zero, which indicates that it manages to successfully solve the task, but miscoordination between the agents occurs frequently.

**Level-Based Foraging:** Because of the large penalty in this environment, Pareto-AC and PACDCG are the only algorithms that achieve returns significantly higher than zero in the LBF task with two agents. We hypothesise that this is a direct outcome of the proposed Pareto-AC objective (Eq. (10)). In the LBF task with three agents, besides Pareto-AC and PACDCG, Friends-QL and MAPPO also manages to achieve a return higher than zero. Pareto-AC achieves returns close to 1 which is the optimal value, while PACDCG, and MAPPO achieve returns close to 0.6 indicating that once the food has a level that requires the cooperation of all three agents they are unable to forage it.

In both LBF and BPUSH tasks, we observe that Pareto-AC significantly outperforms Friends-QL with respect to the achieved returns. This may be caused by the optimal joint policy not being unique, similarly to the Penalty game. Additionally, it has also been observed (Papoudakis et al., 2021) that Q-learning variants achieve lower returns and are less stable than actor-critic algorithms in a variety of environments including grid-worlds. As a result, the differences in the underlying algorithms may be a reason we observe Friends-QL to perform worse than Pareto-AC in terms of achieved returns.

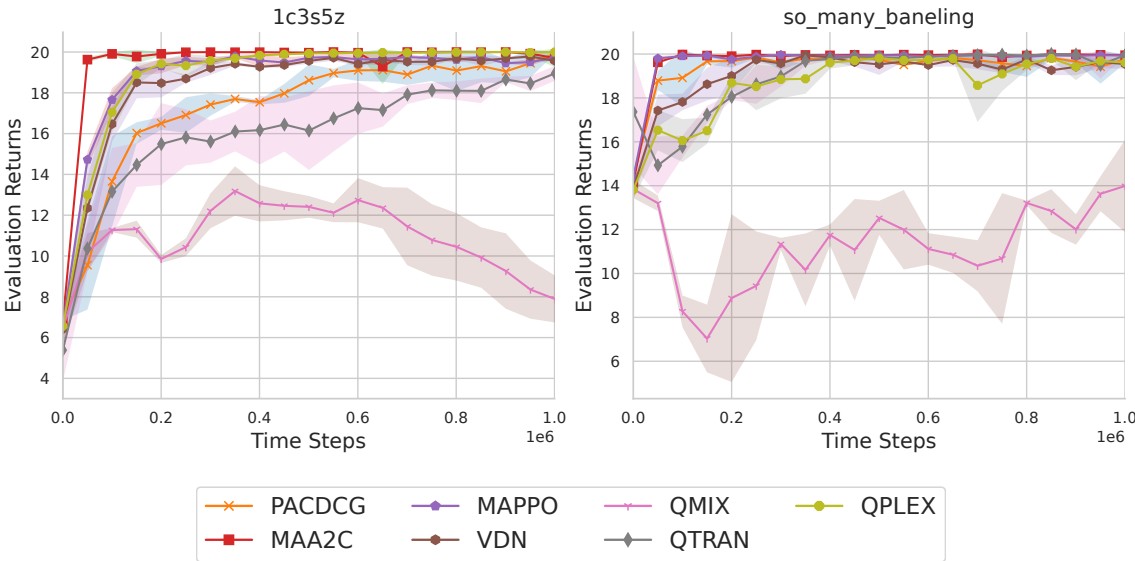

Figure 11: Average episodic evaluation returns and the 95% confidence interval of PACDCG and the evaluated baselines in two SMAC tasks.

### 4.4 Evaluating PACDCG in Environments with Many Agents

The effectiveness of computing $\pi_{-i}^+$ using the DCG critic has also been evaluated in the multi-agent games of Section 4.1. The results presented in Figs. 9 and 10 showed that PACDCG learned in many environments that other algorithms did not. PACDCG converged to the optimal equilibrium in the Climbing (2 agents) and Penalty games and was the second-best algorithm after Pareto-AC in LBF. Therefore, we conclude that the DCG critic can approximate, but not outperform the computation of Eq. (18).

To showcase that PACDCG can be used even in tasks with many agents, where Pareto-AC cannot, we also evaluate in two Starcraft Multi-Agent Challenge (SMAC) tasks. The two tasks, `so_many_baneling` and `1c3s5z`, contain seven and nine agents respectively.

In Fig. 11 we present the returns over the training for PACDCG and all baselines from Section 4.3, except Friends-QL which similarly to Pareto-AC cannot scale to several agents, due to the exponential increase of evaluation that are required for computing the Q-values. The SMAC tasks shown here were selected for their number of agents, and do not exhibit the high-risk-high-reward property, and as such, other algorithms can also solve them. However, this experiment shows the feasibility of computing an approximation of $\pi_{-i}^+$ using value function factorisation algorithms.

That said, high-risk-high-reward settings for large number of agents encounter the problem of exploration. The problem arises when only a few joint actions lead to successful coordination, creating a situation where it becomes difficult to discover these actions within a reasonable number of episodes. As an example, a simple $20 \times 20$ grid-world like LBF (Fig. 8b) with 10 agents could have as many as $10^{26}$ states. If each state has a joint action space of size $4^{10}$, exploring each joint state-action would be completely infeasible. Therefore, the experiments we conducted on many agents were limited to environments in which an informative gradient exists.

### 4.5 Addressing Overestimation and the Effect of N-step Returns

In this section, we explore the use of N-step returns in training the critics of the Pareto-AC algorithm. The critics are responsible for approximating the expected returns for each agent. By incorporating N-step SARSA from the perspective of agent $i$ and Q-learning from the perspective of agents $-i$, we aim to mitigate the overestimation of the expected returns. With N-step returns, Eq. (18) becomes:

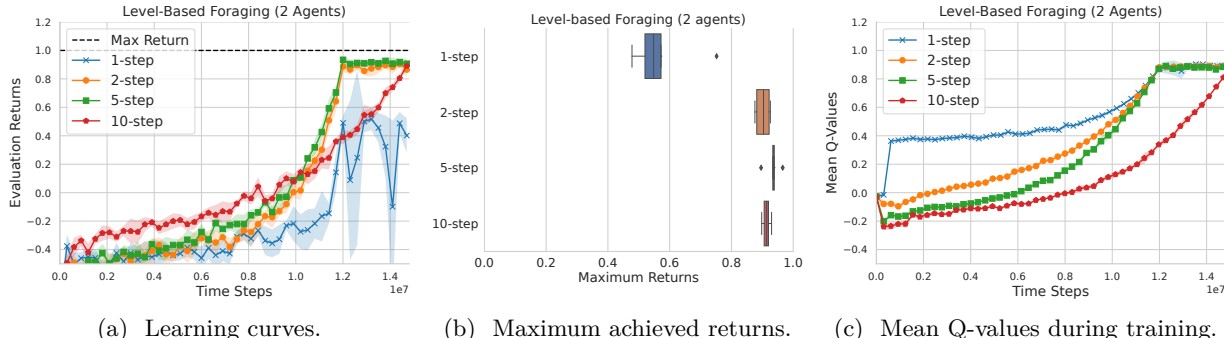

(a) Learning curves.  (b) Maximum achieved returns.  (c) Mean Q-values during training.

Figure 12: Pareto-AC on Level-based Foraging for different values of $N$.

$$\mathcal{L}(\theta_i) = \mathbb{E}_{a_i^t, a_i^{t+1} \sim \pi_i, a_{-i}^t \sim \pi_{-i}} [(\underbrace{r_i^t + \gamma r_i^{t+1} + \ldots + \gamma^{N-1} r_i^{t+N-1}}_{\text{rewards sampled using } \pi} + \gamma^N \underbrace{\max_{a_{-i}} \hat{Q}(s^{t+N}, a_i^{t+N}, a_{-i})}_{\text{approx. } (\pi_i, \pi_{-i}^+) \text{ Q values}} - Q(s^t, a_i^t, a_{-i}^t))^2] \quad (20)$$

The use of N-step returns for training the critic provides a mechanism to balance the returns as sampled from the environment which follows $\pi$, and the approximated returns of $\pi^+$. A large N value reduces the critic to one that simply learns the $Q^\pi$,[3] while when $N = 1$ it approximates the action value function $Q^{\pi^+}$ proposed in Eq. (18). $N$ can be treated as a hyperparameter.

To evaluate the effectiveness of different N values, we conducted experiments with Pareto-AC and N values of 1, 2, 5, and 10 on the LBF environment with two agents. The average evaluation returns over the training are presented in Figure 12a, while the maximum evaluation returns over the five seeds are shown in Fig. 12b. Figure 12c presents the mean Q-values during the course of the training, and we note that the maximum possible *discounted* returns that can be achieved are in the range close to 0.9 – 1.0. The results indicate that:

- $N = 1$ achieves significantly lower returns, because of the overestimated Q-values. The overestimation of the Q-values is clear from Fig. 12c, in which the 1-step line is significantly higher at the early to middle stages of the training than the other $N$ values, even if it does not achieve returns close to that range (see respective returns in Fig. 12a). The overestimation also causes lower converged returns, which is seen in Fig. 12b.

- $N = 2, 5, 10$ do not overestimate their returns and the Q-values (Fig. 12c) closely follow the returns achieved (Fig. 12a). Both $N$ values (and presumably any values in that range) converge to solving the environment with returns close to 1.0.

We conclude that there is an optimal range of $N$ values that is not too narrow and can be easily found with a hyperparameter search (in the case of Level-based Foraging it is values around 5). N values in this range prevent the overestimation of the Q-values while retaining the advantages proposed in Section 3.2.

## 4.6  Evaluation on No-Conflict Matrix Games

The multi-agent games described above are fully cooperative since the reward is common across all agents. However, Pareto-AC works in the general no-conflict case, a superset of the cooperative games. We evaluate our algorithm in the set of all structurally distinct strictly ordinal $2 \times 2$ no-conflict games (Albrecht & Ramamoorthy, 2012; Rapoport, 1966). We train agents with Pareto-AC, Friends-QL, MAA2C, and IQL for

---

[3]Note that $N \to \infty$ does *not* collapse Pareto-AC to A2C as the actor still optimises the modified objective of Eq. (15).

Table 2: Categories of MARL environments.

| Reward Structure | Example Environments | Comments |
|---|---|---|
| General Sum Games | MPE, LBF | Encompasses all subcategories described below. |
| No-Conflict | Stag Hunt, Climbing, SMAC | Strong preference for Pareto-optimal equilibria (see Section 2). |
| Common-Reward | Climbing, Penalty, SMAC, GRF | Subcategory of no-conflict games, widely explored in the literature. |
| Zero-Sum/Competitive | RPS, Chess, Predator-Prey | Pareto optimality has limited significance. |
| Social Dilemma | Prisoner's Dilemma, Harvesting | Pareto optimality is nuanced but relevant. |

five seeds until convergence to one of the joint actions. In those runs, Pareto-AC and Friends-QL solve all 21 games (on all five seeds) by converging to the Pareto-optimal equilibrium, while MAA2C and IQL do not consistently solve three of these games in all five tested seeds. We note that 15 of the 21 games only have one equilibrium, so it is not surprising that every algorithm solved them. The three games that were not solved by other algorithms can be seen in Fig. 13 and exhibit a similar structure to the Stag Hunt (Fig. 1). Note the inclusion of only simpler baselines, since QMIX, VDN, QTRAN and QPLEX assume common-reward games and do not apply to the general no-conflict setting.

Figure 13: Normal-form of the games not solved by MAA2C and IQL. The top-left game is structurally identical to the Stag Hunt (Fig. 1) and the others exhibit a similar structure. All games have two pure Nash equilibria, one that is Pareto-dominated[†] and one that is Pareto-optimal[‡].

# 5    Related Work on Multi-Agent Reinforcement Learning

No-conflict games (Rapoport, 1966; Albrecht & Ramamoorthy, 2012) represent a significant class of multi-agent games. In Section 2 we give a more detailed definition of this game class. A subset of no-conflict games, common-reward games (also referred to as cooperative games) have been studied extensively in the deep MARL community (e.g. Rashid et al., 2018; Foerster et al., 2018; Sunehag et al., 2018; Son et al., 2019; Wang et al., 2021). Pareto-AC naturally applies to this subclass of games, but also to the more general no-conflict setting in which all agents strongly prefer the Pareto-optimal equilibria.

The concept of Pareto equilibrium has been extensively examined in various contexts (e.g. Pardalos et al., 2008; Albrecht et al., 2023). In social dilemma games, such as the Prisoner's Dilemma, achieving Pareto efficiency is a desirable outcome, although it often falls short of addressing the complete issue of finding the best joint policy.[4] Similarly, in other general-sum games, the notion of Pareto efficiency can offer valuable insights. However, in the context of zero-sum games, Pareto optimality is not meaningful, as all joint actions are Pareto optimal. We summarise the main categories of MARL environments based on their reward structure in Table 2.

In this work, we focused on cooperative and no-conflict games, and specifically on environments that contain high-risk-high-reward Nash equilibria. Below, we discuss closely related research topics to Pareto-AC, as well as algorithms that address the equilibrium selection problem and algorithms that fall under the CTDE paradigm.

## 5.1    Equilibrium Selection

Several works have been proposed to address the equilibrium selection problem that mainly focus on accurately learning the Q-value of the joint actions. Littman (2001) proposes two different approaches; one for

---

[4]An example is the classic Prisoner's Dilemma game, where (C, C) yields (-1, -1), (D, C) and (C, D) yield (0, -3) and (-3, 0), respectively, and finally (D, D) yields (-2, -2), the joint action (D, D) represents the only Nash equilibrium but is the sole joint action that does not meet the Pareto efficiency criterion.

games where the other agents are considered friends (Friends-QL), and one for games where the other agents are considered foes (Foe Q-learning). Assuming a game with two agents, in Friends-QL each agent maintains a table of Q-values for all joint actions, which are updated using the following rule:

$$Q_1(s^t, a_1^t, a_2^t) \leftarrow (1 - \alpha)Q_1(s^t, a_1^t, a_2^t) + \alpha(r^t + \gamma \max_{a^{1'}, a^{2'}} Q_1(s^{t+1}, a_{1'}, a_{2'})) \tag{21}$$

where $\alpha$ is the learning rate. From the perspective of each agent $i$, Friends-QL assumes that the other agent $-i$ will always select the action that maximises the joint Q-values of agent $i$. Claus & Boutilier (1998) evaluate independent and joint-action learning in cooperative games with several Nash equilibria, such as the Penalty and the Climbing game. Wang & Sandholm (2002) propose OAL, a joint-action learning algorithm that guarantees convergence to the Pareto-optimal equilibrium. In contrast to Pareto-AC, OAL needs to learn a model of the environment's transition and reward function. More recent works focus on learning the Q-values of the joint actions, by mixing the Q-values for the actions of each individual agent (Sunehag et al., 2018; Rashid et al., 2018; Son et al., 2019), which however are restricted by the representation capacity of the mixing algorithm. Another way of computing the Q-values of the joint actions is based on coordination graphs (e.g. Kok & Vlassis, 2006; Castellini et al., 2019; Böhmer et al., 2020; Wang et al., 2022), where each agent is represented by a node, while the pairwise utility between two agents is represented by an edge, and by using the message-passing algorithm in the graph the Q-value of each joint action can be computed.

Leniency in MARL (Panait et al., 2006) trains agents in a way that they are lenient to their teammates by focusing on the positive outcomes of coordination. Lenient-DQN (Palmer et al., 2018) extends this idea to the deep learning setting by mapping state-action pairs stored in an experience replay to a decaying temperature (Palmer et al., 2018). Similarly, Hysteretic Q-Learning (Matignon et al., 2007) uses two learning rate values in order to reduce the magnitude of negative Q-value updates. Lenient Multi-Agent Reinforcement Learning 2 (LMRL2), introduced by Wei & Luke (2016), employs the mean temperature of an agent's present state and a Boltzmann action selection method to calculate the weight of each action, thereby prioritising exploration in states that have been visited less frequently. All these algorithms attempt to introduce *optimism* (e.g. Claus & Boutilier, 1998; Lauer & Riedmiller, 2000) in multi-agent interaction. Pareto-AC is also optimistic but does so by modifying the learning objective, and is shown to be effective even in POSGs with high-dimensionality, sparse rewards, and significant penalties.

## 5.2 Centralised Training Decentralised Execution

Centralised Training Decentralised Execution (CTDE) is a class of algorithms that mitigate the uncertainty each agent has about the policies of the other agents by allowing information (such as local observations and actions) to be shared during training. During execution, each agent selects their actions in a decentralised manner, only based on their own information (observation-action history). Two large classes of CTDE are the Centralised Policy Gradient Algorithms, and the Value Factorisation Algorithms. Centralised Policy Gradient Algorithms are actor-critic methods that consist of a decentralised actor and a centralised critic. During training, the critic is trained centralised to approximate the joint state (V-value) or joint state-action (Q-value) value function conditioned on the global trajectory of all agents. During execution, the decentralised actor of each agent selects its actions only based on its local trajectory. Some notable centralised policy gradient algorithms are MADDPG (Lowe et al., 2017), COMA (Foerster et al., 2018), MAA2C (Papoudakis et al., 2021) and MAPPO (Yu et al., 2021). Value Function Factorisation Kok & Vlassis (2005) algorithms are limited to cooperative environments, that factorise the global scalar reward into individual utilities for each agent. Such algorithms in the deep learning setting include VDN (Sunehag et al., 2018) and QMIX (Rashid et al., 2018). Both of these approaches require the *monotonicity constraint*, which means that the argmax of the joint Q-value is the same as the joint action that is generated if we perform the argmax operator in the individual Q-values of each agent. There are several algorithm variations in both categories that extend that focus on relaxing this constrain to be able to approximate a broader class of joint Q-values, such as QTRAN (Son et al., 2019), and QPLEX (Wang et al., 2021), which however do not always yield higher returns compared to the simpler VDN, and QMIX algorithms.

## 6 Conclusion

We proposed Pareto-AC, an actor-critic MARL algorithm designed to converge to Pareto-optimal Nash equilibria in no-conflict multi-agent games. The algorithmic choices behind Pareto-AC are based on the assumption that in no-conflict games, each agent can assume that the other agents will choose the action that will lead to the Pareto-optimal equilibrium. While the community has developed approaches to solve matrix games with risk-averse equilibria, this work extends these approaches to the deep learning setting and environments with high dimensionality. Our results showed that Pareto-AC is able to achieve higher returns, and converges more often to the Pareto-optimal equilibrium, compared to several state-of-the-art MARL algorithms, such as MAPPO and QPLEX. Pareto-AC also outperforms a deep learning implementation of the Friends-QL algorithm since it learns to converge to one of the multiple equivalent equilibria using its policy network. Additionally, to address the computational bottleneck that is raised by the exponentially increasing number of Q-values, we proposed PACDCG as a scalable approximation of Pareto-AC that uses the DCG algorithm for its critic. Two main directions of future research emanate from this work. The first direction is the theoretical understanding of Pareto-AC, and under which requirements we can guarantee that it will converge to a Nash equilibrium or even a Pareto optimal Nash equilibrium. Second, we aim to explore extensions beyond no-conflict games, replacing the max operator with min-max based approaches (e.g. Littman, 2001) for zero-sum games.

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

# A   Implementation and Hyperparameter Details

Throughout the hyperparameter search, we systematically examined multiple configurations for the training process for *both the baseline algorithms and Pareto-AC*. Our approach ensured fairness by maintaining a roughly equal number of search configurations for all algorithms under consideration. This included testing hidden dimensions of 64 and 128, learning rates of 0.0003 and 0.0005, considering both Fully Connected (FC) and GRU network architectures, and experimenting with initial entropy coefficients of 0.1, 0.8, 4, and 20, as well as final entropy coefficients of 0.001, 0.01, and 0.02 (entropy only applies to PG algorithms). To select the hyperparameter configurations used to generate the results (Tables 3 to 5) that were presented in Section 4, we trained all hyperparameter configurations for five different seeds for each algorithm. Then we chose the configuration that achieves the maximum evaluation return averaged over all evaluation time steps and seeds.

The norm of the gradient is clipped to 10 in all algorithms. In the actor-critic methods, we subtract the policy's entropy, multiplied by an entropy coefficient, from the policy gradient loss to ensure sufficient exploration. We found it beneficial to start with a large entropy coefficient and gradually reduce it to a smaller value throughout training. We set initial entropy values to 1.5 and 0.8, and final entropy coefficient values to 0.01, 0.02, and 0.03.

Table 3: Hyperparameters for Pareto-AC in all environments.

|                      | Climbing | Penalty | Climbing 2 Ag. | Boulder Push | LBF 2 Ag. | LBF 3 Ag. |
|----------------------|----------|---------|----------------|--------------|-----------|-----------|
| hidden dimension     | 64       | 64      | 64             | 128          | 128       | 128       |
| initial entropy coeff| 4        | 4       | 20             | 0.8          | 0.8       | 0.8       |
| final entropy coeff  | 0.1      | 0.1     | 0.1            | 0.01         | 0.02      | 0.02      |
| entropy anneal perc  | 80%      | 80%     | 80%            | 100%         | 80%       | 100%      |
| learning rate        | 0.0003   | 0.0003  | 0.0003         | 0.0005       | 0.0003    | 0.0003    |
| network type         | FC       | FC      | FC             | GRU          | FC        | FC        |

Table 4: Hyperparameters for PACDCG in all environments.

|                      | Climbing | Penalty | Climbing 2 Ag. | Boulder Push | LBF 2 Ag. | LBF 3 Ag. |
|----------------------|----------|---------|----------------|--------------|-----------|-----------|
| hidden dimension     | 64       | 64      | 64             | 128          | 64        | 64        |
| initial entropy coeff| 4        | 4       | 20             | 0.8          | 1.5       | 1.5       |
| final entropy coeff  | 0.1      | 0.1     | 0.02           | 0.01         | 0.03      | 0.02      |
| entropy anneal perc  | 80%      | 80%     | 80%            | 100%         | 100%      | 100%      |
| learning rate        | 0.0003   | 0.0003  | 0.0003         | 0.0005       | 0.0003    | 0.0003    |
| network type         | FC       | FC      | FC             | GRU          | FC        | FC        |

Table 5: Hyperparameters for PACDCG in the SMAC tasks.

|                  | so_many_baneling | 1c3s5z |
|------------------|------------------|--------|
| hidden dimension | 64               | 64     |
| entropy coeff    | 0.01             | 0.01   |
| learning rate    | 0.0005           | 0.0003 |
| network type     | FC               | FC     |

