# OpenReview forum: "Pareto Actor-Critic for Equilibrium Selection in Multi-Agent Reinforcement Learning"
_TMLR — Accepted by TMLR_

### Review · Reviewer_t3RA · 2023-07-23

**Summary Of Contributions:**

This paper studies selecting Pareto-optimal equilibrium in non-conflict games through deep multiagent reinforcement learning. The authors proposed Pareto Actor-Critic, a deep MARL algorithm where every player's objective function assumes opponents always maximize for its own utilities. The authors derived the policy gradient update, where it learns an approximate joint Q function to derive opponents' lenient actions. To address the exponential cost when selecting the opponents' joint action the authors proposed learning a deep coordination graph to decompose the computation cost. The author presents empirical results where PAC outperforms baseline algorithms in reaching Pareto-optimal equilibria.

**Audience:**

Yes

**Claims And Evidence:**

Yes

**Requested Changes:**

Some questions that I hope the authors can address:

(1) Can the author elaborate more on the objective function (11). Why the trajectory distribution is on-policy while the values are assumed to be the optimistic one. Can one switch them and how would that affect the approach.

(2) Why maximizing the per-timestep joint action of the opponents in equation (17) optimizing the objective (11). Is it possible to derive a convergence result based on Markovian properties.

(3) An MARL algorithm that chooses opponents' joint action to maximize its own Q-value sounds familiar to me. Can the authors compare your approach to the ones in the following papers.
[1] The Dynamics of Reinforcement Learning in Cooperative Multiagent Systems. Claus & Boutiiler
[2] Reinforcement Learning to Play an Optimal Nash Equilibrium in Team Markov Games. Wang & Sandholm

(4) Is it possible to generalize your approach to domains beyond no-conflict games? And how should you modify the algorithms to account for more general domains?

**Strengths And Weaknesses:**

Strengths: The problem is well-motivated. The presentation is very clear. The approach is sound. Empirical results are fairly strong.

Weaknesses: It is somehow restricted to no-conflict games. Some theoretical justification are missing.

---

> ### Author Response · Authors · 2023-09-10
> **Author Rebuttal**
>
> We would like to thank the reviewer for their time and feedback. Please see below for our reply to their comments.
>
> > (1) Can the author elaborate more on the objective function (11). Why the trajectory distribution is on-policy while the values are assumed to be the optimistic one. Can one switch them and how would that affect the approach.
>
> The trajectories are gathered under \pi while the value is under the policy $\pi^+$. This is a standard instantiation of the off-policy actor-critic algorithm [Degris et al 2012]. One could potentially also gather optimistic samples, but this would require training each agent in a parallel environment, where the agent is following its policy and the other agents are using the optimistic policy, which would significantly increase the computational burden.
>
> > (2) Why maximizing the per-timestep joint action of the opponents in equation (17) optimizing the objective (11). Is it possible to derive a convergence result based on Markovian properties.
>
> Equation 11 is the optimisation objective where the policy $\pi$  is used to gather the training trajectories, and the value function under the optimistic policy $\pi^+$ is optimised. Equation 17 shows how we estimate the optimistic policy for agents $-i$. By optimising the objective in Equation 11, the policy of agent $i$ will empirically converge to the Pareto-optimal policy, because this is the best response to the policy estimated by Equation 17, and this will happen for every agent $i$, leading the joint policy to be Pareto-optimal.
> We have added the need for a more concrete theoretical justification as a future work direction in the conclusion.
>
> > (3) An MARL algorithm that chooses opponents' joint action to maximize its own Q-value sounds familiar to me. Can the authors compare your approach to the ones in the following papers. [1] The Dynamics of Reinforcement Learning in Cooperative Multiagent Systems. Claus & Boutiiler [2] Reinforcement Learning to Play an Optimal Nash Equilibrium in Team Markov Games. Wang & Sandholm
>
> The closest match for our work is Friends-Q Learning by Michael Littman, which we discuss and compare against. We believe that a large part of our contribution is to bring the idea of considering joint actions (e.g. Friends-QL was only implemented in tabular settings) into the deep-learning setting. The papers that the reviewer proposes explore similar ideas but generally require to learn models either of the other agents, or the environment. Comparison between these types of algorithms with model-free MARL is not straightforward (especially in high-dimensionality environments), and for this reason, we selected Friends-QL as a more appropriate baseline. That said, we now discuss these papers in the related work section.
>
> > (4) Is it possible to generalize your approach to domains beyond no-conflict games? And how should you modify the algorithms to account for more general domains?
>
> Pareto optimality is not necessarily desired in all types of games. E.g. in zero-sum games all joint policies are Pareto-optimal. One could potentially compute any equilibrium of interest from the joint-Q value of each agent and use it in the advantage estimation. Still, we would like to note that no-conflict games are a large class of games, *larger* than the typical cooperative (common-reward) class that has been studied a lot by the community. We have added text in Sec. 5 that discusses other games in the context of Pareto-AC.

---

### Review · Reviewer_oaYN · 2023-08-16

**Summary Of Contributions:**

The paper considers the problem of equilibrium selection in multi agent games, seeking a Pareto optimal equilibrium among the set of possible equilibria. Many existing MARL algorithms may not converge or converge to a low social welfare outcome as agents are uncertain about the policies of other players. The authors propose an approach called Pareto Actor Critic (ParetoAC) which is a variant on traditional actor critic RL algorithms, that is tailored to no-conflict games (which contains cooperative games, but is not as restrictive), and aims at maximizing social welfare, or converging to a Pareto-optimal equilibrium.

The empirical analysis in the paper considers several interesting multi agent games, and shows that the proposed approach achieves higher total returns than existing known MARL approaches, and shows convergence to a Pareto optimal equilibrium in various matrix games. The authors also show a GNN based extension, which scales well to games with many agents.



**Audience:**

Yes

**Broader Impact Concerns:**

No substantive concerns that I can see.

I would mention in the paper that Cooperative AI where agents collaborate with one another could be used for bad purposes, e.g. consider a set of agents who are cooperative to collude in an auction. But this concern relates to any work on cooperating agents.

**Claims And Evidence:**

Yes

**Requested Changes:**

First, it would be good to include a more in depth discussion of MARL as a whole. For instance, it would be good to include a table with types of games (competitve, two player zero sum, many player zero sum, social dilemmas, no-conflict games and cooperative games). For each of those, a few key games falling in this class could be mentioned, as well as results indicating convergence to sub-optimal outcomes (for multiple of the baselines that you have mentioned).

Also, it would be good to include a more detailed high level diagram of the method that you proposed, and the key building blocks, early in the paper. The algorithm is brough in full a bit later, but an early introduction would be welcome.

In terms of the empirical results, I think a slight extension / discussion of other known results on MARL in Cooperative settings would be very welcome, as well as trying several know semi-coopertive games. One example is congestion games, where agents attempt to avoid selecting the same resources (as congested resources are more costly to use - see Wikipedia), and another example might be an extension of boulder push where the are multiple boulders (say 3) and multiple agents (say 6), so agents have to partition into teams (or possibly where only one boulder may be moved at the same time). As is, the empirical results are for somewhat limited settings.

Finally, you examine a non-cooperative game theoretic solution (e.g. Nash). I think it would also be interesting to comment on Cooperative game theory / coalitional game theory solutions (e.g. the Core) - is there anything you can say here about RL being used for team formation or ad-hoc teamwork in these environments. Similarly, the algorithm was proposed for cooperative environment, but it would be interesting to examine what it does in mixed motive settings - could you comment on that?

All in all, this is very interesting work, and I think this would likely be pursued by further researchers examining more cooperative games. Currently there is a lot of work on competitive settings or mixed-motive ones, but less on cooperative games.


**Strengths And Weaknesses:**

I really love the topic of the paper, which is considering (semi-) cooperative games, and identifying algorithms that are likely to converge to a high welfare equilibrium. The authors also do a good job illustrating the issues arising with exploration and non-stationarity that result in converging to possibly low outcome results.

The empirical analysis is on relatively few games, and with reasonable MARL baselines, but is quite illuminating.

I think the paper could benefit from extending the empirical results a bit, and also from improving on the presentation and the writing. See below.

---

> ### Author Response · Authors · 2023-09-10
> **Author Rebuttal**
>
> We would like to thank the reviewer for their time and feedback. Please see below for detailed replies to the comments.
>
> > A table with types of games
>
> We now briefly discuss the main categories of multi-agent games and summarise them in a table (Sec. 5).
>
> > high-level diagram of the method that you proposed
>
> While Figures 3 and 4 offer high-level descriptions, we agree that a more complete diagram early on can be useful. We have now included a high-level diagram much earlier, in Section 3 (page 4).
>
> > Limited empirical results
>
> Pareto-AC aims to train agents that learn to converge to a Pareto-optimal (Nash) equilibrium in high-risk-high-reward no-conflict games. Many studied MARL environments (e.g. MPE, GRF, overcooked, etc.) have dense reward functions to assist the learning process in finding the optimal joint policy, and do not fall under the category of games that Pareto-AC targets (no-conflict, suboptimal equilibria). Our aim is to keep our experiments and analysis focused on the type of games that present the specific characteristics.
>
> In terms of congestion games (https://en.wikipedia.org/wiki/Congestion_game), they do not fall under the no-conflict class of games (note that the tables on the Wikipedia page refer to costs, so players want them to be smaller). The games where agents are partitioned into teams are also very interesting, but Pareto-AC does not necessarily address this issue (maybe some future work can!) and therefore such experiments might not provide any useful conclusion. In the new text and table in Sec. 5 we discuss the many types of MARL games in the context of Pareto-AC.
>
>
> > Finally, you examine a non-cooperative game theoretic solution (e.g. Nash). I think it would also be interesting to comment on Cooperative game theory / coalitional game theory solutions (e.g. the Core) - is there anything you can say here about RL being used for team formation or ad-hoc teamwork in these environments?
>
> Generally, we are concerned with Partially Observable Stochastic Games, in which the reward function is predefined and the rewards are always distributed according to it. In common-reward games, this would mean that the reward is always split equally, and therefore the setting is equivalent to a "forced" grand coalition. This is relaxed in a no-conflict game, in which an agent can choose not to participate (but the setting does not allow agents to choose not to cooperate and not share the rewards). Cooperative game theory studies how to fairly distribute rewards to form stable coalitions, so a change of the setting would be required in which the agents can decide (possibly beforehand) with whom to share the rewards. Alternatively, the agents can aim to approximate a different reward (e.g. Shapley value) but it is not clear to us if this would translate to improved performance on the POSG system.
>
> > Similarly, the algorithm was proposed for a cooperative environment, but it would be interesting to examine what it does in mixed-motive settings - could you comment on that?
>
> In mixed-motive games, it is harder to justify the use of Pareto optimality (in the extreme case of zero-sum games, all policies are Pareto-optimal). However, in cases where a Pareto-optimal joint policy is acceptable as a solution for the game, Pareto-AC will not be able to reliably converge to the Pareto equilibrium, as the agents will still try to maximise their own rewards. This is now briefly discussed in the new text in Sec 5.

---

### Review · Reviewer_q1VR · 2023-08-27

**Summary Of Contributions:**

This paper introduces Pareto Actor-Critic (PAC), an actor-critic algorithm that is specialized in no-conflict games, where an agent maximizes the utility of all agents by maximizing its own utility. Other Reinforcement Learning algorithms were shown to not necessarily converge to a Pareto optimal equilibrium in this type of game. PAC, on the other hand, was shown to converge (or very close to converge) to the Pareto optimal equilibrium on all games evaluated.

The trick used to achieve such a result is to assume that the other players, denoted $-i$ follow a policy that maximizes player's $i$ utility. Intuitively, this is telling player $i$ that they should trust that the other players are also cooperative. While this will incentivize $i$ to converge to a Pareto optimal equilibrium, it will in turn also incentivize the other players to converge to the same equilibrium.

PAC is also extended with a Deep Coordination Graph (DCG) to handle domains with many agents, as the number of actions grows exponentially with the number of players. The DCG is used to approximate the argmax operator used in Q-Learning.

The paper presents experiments on a variety of domains, including simple but informative normal-form games, grid-world domains, and two challenging problem domains from Starcraft. The results support the claims that PAC and PACDCG are able to converge to Pareto optimum equilibria.

**Audience:**

Yes

**Broader Impact Concerns:**

I don't have any broader impact concerns with this submission.

**Claims And Evidence:**

Yes

**Requested Changes:**

I would like the authors to handle all my suggestions related to the presentation of the paper. While it would be nice to see experiments on more challenging no-conflict games, this is not a requirement for me to recommend acceptance. The claims the authors made are already backed up by empirical evidence.

I also would like to see an in-depth discussion on the reasons why PAC is able to converge on some of the domains where Friends-QL isn't.

**Strengths And Weaknesses:**

**Strengths**

Almost everything in this paper makes sense: the design decisions, the choice of problem domains, the experiments, and the claims made. The paper is mostly easy to follow and makes a reasonable contribution in the context of no-conflict games. I absolutely love the intuition provided in the first paragraph of page 2. Well done!

**Weaknesses**

The paper boils down to a comparison of Actor-Critic algorithms and Q-Learning algorithms in the context of no-conflict games through PAC and Friends-QL. While this can be perceived as a modest contribution, I appreciate that the comparison was nicely done. However, I still don't fully understand why PAC is able to converge in some of the games while Friends-QL isn't. An in-depth discussion on this matter would be valuable. My primary concern is that this observed discrepancy may simply stem from insufficient parameter tuning for Friends-QL, rather than any inherent limitations or advantages of the algorithms themselves.

Since this is an empirical paper, I would expect evaluations on more challenging no-conflict problems. The only very challenging domains used in the evaluation are the Starcraft ones, which are adversarial.

I would expect the paper to better explain a few core concepts used, so the reader could better understand the approach without having to rely on other papers. In particular, I have the following comments about presentation:

1. An example of DCG would have helped answering some lingering questions, such as "Why does Equation 19 can approximate the joint Q-value?" "How is a message passing algorithm useful at all?"
2. What are the input format used to train the neural model in each domain?
3. What is the target network $\hat{Q}$ used in Equation 18? If $\hat{Q}$ is the target network, then what is $Q$?
4. Algorithm 1 doesn't have inputs nor outputs, which makes it confusing to understand exactly what it does.
5. Why is it an approximation in Equation 12?
6. Why does inequality 14 hold true?
7. What is the sum in Equation 11 over? All states in the state space?
8. What is the expectation over at the bottom of page 2?
9. Why is $\pi_{a_{-i}}^+(a_{-i}|s, a_i)$ in Equation 12 conditioned on $a_i$?
10. The plots aren't colorblind friendly. This is not acceptable.

Other minor comments with respect to presentation:

1. The sentence "There is no incentive for any of the agents to deviate from that equilibrium" is awkward because this is the very definition of an equilibrium.
2. In the caption of Figure 2(b) should read $\pi_2^+(B) = 0$ instead of $\pi_2^+(A) = 0$.
3. The design choices shown in Figure 3 doesn't address the issue with the exponential growth of the number of actions, but only mitigates it. The last sentence in the third paragraph of Section 3.3 needs rewording.
4. The paper should cite SARSA in the paragraph above the pseudocode.
5. The paragraph describing the Level-Based Foraging is quite rough, please review it carefully.

---

> ### Author Response · Authors · 2023-09-10
> **Author Rebuttal (Part I)**
>
> We would like to thank the reviewer for their time and feedback. Please see below for our reply to individual comments.
>
> > **PAC vs FQL:**
>
> We do not think that the differences can be attributed to hyperparameter search because 1) we observe similar results in the "easy" matrix games (especially the Penalty game - refer to Fig 8) where Pareto-AC converged to a Pareto-optimal (Nash) equilibrium across a variety of hyperparameters, while FriendsQL did not; and 2) we performed a "fair" hyperparameter search for all our baselines, i.e. the number of tested combinations was high and roughly equal across all algorithms (details of the hyperparameter search are now clearly stated in Sec 4.2 and the appendix).
>
> We believe the clue lies in the Penalty game, which FriendsQL cannot solve reliably (Fig **8**): the Penalty game has *two* Pareto-optimal equilibria (instead of a single one as in the Climbing game). The joint action-value function of FriendsQL captures the values of the joint actions, but extracting a policy requires an argmax operation, which returns a set of actions, and any ties are broken arbitrarily (in deep RL, value approximations are noisy enough to produce the required randomness). To illustrate this, let us return to the Penalty game where both (A,C) and (C,A) are optimal and reward agents with (10). Even if FriendsQL approximates (*almost*) perfectly the joint action values, when selecting the actions in a decentralised way, actions (A,A) or (C, C) might still be performed as the respective values for individual actions A and C are approximately equal, therefore heavily penalising the agents.
>
> A policy gradient method does not have this problem, as during the training process if agent 1 assigns a higher probability say to action A compared to action C, agent 2 will be strongly reinforced to converge to the respective action (C in this case) and avoid the heavily-penalised A action (and the opposite direction also holds).
>
> Of course, the Penalty game is a toy example, but higher-dimensionality games can have very similar dynamics as the optimal joint policy is rarely unique. We hypothesize that what we discussed above causes similar issues in these games and is detrimental to learning.
>
> That said, DQN (and variants) has been observed to be less successful and stable than PGs in a variety of environments including grid-worlds [see Papoudakis et al. 2021], which means that the differences in the underlying algorithms can be a further reason we observe FriendsQL to be worse than Pareto AC.
>
> We have included a refined version of this discussion in the results section (4.3).
>
> > **Only SMAC is challenging**
>
> We would like to respectfully argue that SMAC is not the only challenging environment. For example, Papoudakis et al 2021 have shown that several gridworld environments, including LBF, can be more challenging than SMAC in the sense that few algorithms are able to achieve the optimal return. Indeed, as we show, all included baselines (which include some very popular MARL algorithms) do not solve the tested version of LBF, or perform suboptimally in BoulderPush, which is further evidence that these are indeed very challenging environments.
>
> We would also like to note that SMAC is also not an adversarial environment. While there is an opponent team, it is controlled by the game's controller, and therefore, to the best of our knowledge, throughout the literature it is assumed to be part of the environment.

---

> ### Author Response · Authors · 2023-09-10
> **Author Rebuttal (Part II)**
>
> We have also addressed the reviewer's concerns regarding the presentation issues. A detailed list of changes/replies (where required) can be found below:
>
> 1. DCG is trained using TD-learning to approximate the joint Q-value. DCG is an extension of VDN, with the main difference between the two of them being that DCG also models the pairwise utilities between the agents.
> As Bohmer et al note, if the edges are dropped in DCG, it collapses to VDN.
> The message-passing algorithm is used to estimate the messages and the individual actions of the agents that maximise the joint Q-value. We have added a brief discussion of this in 3.4.
>
> 2. The input to the network of each agent is a vector-based observation that contains the relative distances of the agent from other agents and objects in the grid-world, that exist within its perceptive field. We have included the observation space in Sec 4.1 which describes the environments.
>
> 3. Q is the Q-network parameterised by theta, that approximates the Q-value. \hat{Q} is the target network as now described right under Eq 18.
>
> 4. Thank you for letting us know. This has now been addressed.
>
> 5. This is the approximation based on the work of Degris et al 2012. The Q-value has also a gradient with respect to the parameters. However, in Theorems 1 and 2,  Degris et al have shown that the term can be omitted and still minimise the loss function. We have added a short sentence describing this in the paper (equation 5 of Degris et al 2012).
>
> 6. That is another result from Degris et al. 2012 (Theorem 1), which shows that if the policy is represented in a tabular form, following this direction optimises the state value function of all states.
>
> 7. Yes, it is a sum over all states. This is now more clearly stated in Eq 11 under the summation.
>
> 8. It is an expectation under the actions that are generated from the joint policy, and the states that are generated from the state transition function. We have added a clarification in Section 2.1.
>
> 9. This is by (our) definition, as finding the action a_{-i} that optimises the joint Q-value needs to know what a_i is. (also see Eq 9 and 17): At each time step, the Pareto policy of the agents -i define the actions that maximise the joint Q-value of agent $i$ given the action of agent i.
>
> 10. Thank you, markers have been added to the line plots.
>
> Minor comments
> Thank you for bringing these to our attention. These have been now fixed.
>
> References
>
> Georgios Papoudakis, Filippos Christianos, Lukas Schäfer, and Stefano V Albrecht. Benchmarking multiagent deep reinforcement learning algorithms in cooperative tasks. In Conference on Neural Information Processing Systems Datasets and Benchmarks Track, 2021.
>
> Thomas Degris, Martha White, and Richard S Sutton. Off-policy actor-critic. In International Conference on Machine Learning, 2012.
>
> Wendelin Böhmer, Vitaly Kurin, and Shimon Whiteson. Deep coordination graphs. In International Conference on Machine Learning, 2020.

---

### Author Response · Authors · 2023-09-10
**Author Rebuttal**

We would like to thank the reviewers for their valuable time and useful feedback. We have prepared a response for each point raised by the reviewers. We have also updated the submission with the amendments mentioned in the rebuttal responses.
Please let us know whether there are additional comments from the reviewers.

---

### Decision · Action_Editors · 2023-10-05

**Recommendation:** Accept as is

**Comment:**

All of the major points brought up in review appear to have been addressed by the authors. However, please double-check to ensure that all of the reviewers' points have been addressed and upload a new version, if necessary.

There are no requirements to make changes, but please add author names and affiliations for the camera-ready copy.

**Audience:**

Cooperative MARL is probably the largest subdivision of MARL, often appealing to the single-agent RL crowd as well. The audience is certainly appropriate for TMLR.

**Claims And Evidence:**

The paper presents a Pareto Actor-Critic algorithm, a multiagent RL algorithm which resolves the equilibrium selection problem in no-conflict games. The paper demonstrates that the algorithm does indeed finds the Pareto-optimal equilibrium in matrix games, with an impressive empirical demonstration comparing to seven other MARL algorithms. The algorithm also has an approximate neural form which is applied to multiagent grid worlds commonly used in the MARL literature.

The claims are substantiated by evidence shown in the paper.